# Predictive Fermentation Control of *Lactiplantibacillus plantarum* Using Deep Learning Convolutional Neural Networks

**DOI:** 10.3390/microorganisms13112601

**Published:** 2025-11-15

**Authors:** Chien-Chang Wu, Jung-Sheng Chen, Yu-Ching Lu, Jain-Shing Wu, Yu-Fen Huang, Chien-Sen Liao

**Affiliations:** 1Institute of Biotechnology and Chemical Engineering, I-Shou University, Kaohsiung 840203, Taiwan; 2Department of Medical Research, E-Da Hospital, I-Shou University, Kaohsiung 824005, Taiwan; 3Division of Respiratory Medicine, Department of Internal Medicine, E-Da Hospital, I-Shou University, Kaohsiung 824005, Taiwan; 4Department of Computer Science and Information Management, Providence University, Taichung 433303, Taiwan; 5Institute of Biopharmaceutical Sciences, National Sun Yat-sen University, Kaohsiung 804201, Taiwan; 6Department of Medical Science and Biotechnology, I-Shou University, Kaohsiung 824005, Taiwan; 7Graduate School of Human Life and Ecology, Osaka Metropolitan University, Osaka 5588585, Japan

**Keywords:** deep learning, fermentation prediction, *Lactiplantibacillus plantarum*, convolutional neural network, bioprocess control, probiotics

## Abstract

The fermentation of *Lactiplantibacillus plantarum* is a complex bioprocess due to the nonlinear and dynamic nature of microbial growth. Traditional monitoring methods often fail to provide early and actionable insights into fermentation outcomes. This study proposes a deep learning-based predictive system using convolutional neural networks (CNNs) to classify fermentation trajectories and anticipate final cell counts based on the first 24 h of process data. A total of 52 fermentation runs were conducted, during which real-time parameters, including pH, temperature, and dissolved oxygen, were continuously recorded and transformed into time-series feature vectors. After rigorous preprocessing and feature selection, the CNN was trained to classify fermentation outcomes into three categories: successful, semi-successful, and failed batches. The model achieved a classification accuracy of 97.87%, outperforming benchmark models such as LSTM and XGBoost. Validation experiments demonstrated the model’s practical utility: early predictions enabled timely manual interventions that effectively prevented batch failures or improved suboptimal fermentations. These findings suggest that deep learning provides a robust and scalable framework for real-time fermentation control, with significant implications for enhancing efficiency and reducing costs in industrial probiotic production.

## 1. Introduction

Microbial fermentation has become a cornerstone of modern biotechnology, enabling the production of probiotics, enzymes, amino acids, bioactive peptides, and pharmaceuticals on an industrial scale. The global microbial fermentation technology market is projected to surpass USD 36 billion by 2025, reflecting both rising consumer demand and expanding industrial applications [1]. Among probiotic strains, *L. plantarum* has emerged as one of the most widely utilized species in functional foods and nutraceuticals because of its strong acid tolerance, broad metabolic versatility, and health-promoting effects [2,3,4]. Despite its industrial significance, the fermentation of *L. plantarum* remains challenging to standardize due to nonlinear interactions among multiple parameters such as pH, temperature, inoculum density, and medium composition. This inherent complexity often results in poor reproducibility, inconsistent yields, and financial losses for manufacturers.

Conventional strategies for fermentation monitoring, such as offline sampling, endpoint CFU measurements, or physicochemical assessments, are inherently reactive and frequently fail to capture the dynamic evolution of microbial growth [5]. Although statistical approaches like Response Surface Methodology (RSM) have been applied to optimize culture conditions, they remain limited by assumptions of linearity and cannot adequately model the complex temporal behavior of microbial systems [6]. Consequently, there is an urgent need for advanced predictive tools that can provide early, actionable insights to prevent failure and maximize productivity.

Recent advances in artificial intelligence (AI) have demonstrated significant potential in biological process modeling. Artificial Neural Networks (ANNs) have been employed to optimize exopolysaccharide production [7], predict GABA synthesis in soymilk fermentation [8], and enhance cellulase yields under solid-state fermentation [9]. Hybrid optimization strategies, such as combining ANN with Genetic Algorithms (GA) or RSM, have further improved multi-variable optimization of lactic acid bacteria processes [10,11]. More recently, AI-driven approaches have extended to morphological classification of microbial images [12], the optimization of recombinant protein production [13], and even predictive models for food safety applications [14]. These studies collectively highlight the versatility of AI as a transformative tool in fermentation science.

Convolutional neural networks (CNNs), originally developed for image recognition, have proven particularly adept at extracting patterns from high-dimensional and structured data [15]. Their ability to capture local dependencies and temporal trends makes them highly suitable for analyzing fermentation time-series data. Preliminary studies have shown that CNNs outperform traditional statistical models and even recurrent neural networks in discriminating microbial growth trajectories and predicting process outcomes [9,16,17,18]. This suggests that CNNs may serve as effective “soft sensors” for bioprocess control, capable of delivering accurate early-stage predictions that traditional models cannot provide.

Despite these promising advances, most existing AI applications in fermentation rely on complete fermentation profiles or final-stage data, limiting their practical utility in real-time decision-making. Few studies have systematically investigated the use of CNNs for early-stage prediction of probiotic fermentation success and the implementation of timely corrective interventions. In particular, there is a lack of integrative frameworks that combine predictive modeling with experimental validation in industrially relevant probiotic strains such as *L. plantarum*.

In this study, we propose a CNN-based predictive framework that utilizes the first 24 h of *L. plantarum* fermentation data to classify process outcomes into successful, semi-successful, or failed categories. By integrating feature selection, time-series modeling, and validation experiments, we demonstrate the capacity of CNNs to provide robust early-stage predictions and enable effective real-time interventions. This work contributes a novel, scalable, and intelligent approach for probiotic fermentation control, with the potential to improve product consistency, reduce batch failure, and enhance overall industrial efficiency.

## 2. Materials and Methods

### 2.1. Bacterial Strain and Culture Conditions

The strain *L. plantarum* LP198 was obtained in lyophilized powder form from SYNBIO TECH, Inc. (Kaohsiung, Taiwan). The strain was rehydrated and activated in de Man–Rogosa-Sharpe (MRS) medium (Cyrus Bioscience, Taipei, Taiwan) at 37 °C for 48 h, and then subcultured at a 10% (*v*/*v*) inoculum into fresh MRS medium to generate sufficient biomass for fermentation. Viability was confirmed by colony-forming unit (CFU) enumeration using plate counts on MRS agar, according to the CNS 14760 N6371 protocol for lactic acid bacteria [17]. Long-term preservation was achieved by storage in 65% glycerol at −80 °C or under manufacturer-recommended lyophilized conditions, with activation steps performed prior to each experiment to ensure cell viability and consistency. As illustrated in Figure 1, this strain activation and culture process constituted the initial stage of the workflow, which subsequently integrated fermentation setup, data acquisition, preprocessing, and deep learning prediction.

### 2.2. Fermentation Setup and Data Acquisition

Fermentation experiments were conducted in 12 L bench-top bioreactors (BTF-A, Biotop, Nantou, Taiwan) with a working volume of 10 L, operated under batch conditions. Each vessel was equipped with embedded pH, temperature, and dissolved oxygen (DO) probes and connected to a computer-based data acquisition system for continuous signal logging. The fermentation process was maintained at 37 °C, with agitation at 200 rpm and an initial pH adjusted to 6.5. Activated seed cultures of *L. plantarum* LP198 were inoculated at 10% (*v*/*v*), corresponding to approximately 1 × 10^7^ CFU/mL at the start of fermentation. Dissolved oxygen, pH, and temperature were automatically monitored at 5 min intervals throughout the 48 h fermentation period.

Outlier values exceeding three standard deviations from the mean were removed, and missing values were linearly interpolated based on adjacent time points. To focus on early-stage predictive features, all datasets were truncated to the first 24 h (288 time points). Subsequently, the cleaned data were normalized to zero mean and unit variance to ensure comparability across runs. As shown in Figure 2, the preprocessing pipeline comprised raw signal acquisition, outlier removal, interpolation, truncation to 24 h, and normalization for subsequent model training.

Software and data management. All bioprocess signals were digitally logged through the BTF-A controller (Biotop, Taiwan) and exported as comma-separated value (CSV) files for downstream analysis. Data management and preprocessing, including outlier filtering, linear interpolation, 24 h truncation, and normalization, were conducted in Python (v3.9) (Anaconda distribution) using the Pandas (v2.2.3), NumPy (v1.23.0), and SciPy libraries (v1.16.2), with visualization performed in Matplotlib (v3.3.2). Feature ranking and statistical utilities were implemented in scikit-learn (v1.2.2), and model development was carried out using TensorFlow/Keras (v2.12) as described in Section 2.5. These tools collectively ensured standardized data handling, transparency, and full reproducibility of the analytical workflow.

### 2.3. Dataset Labeling and Classification Criteria

For each fermentation run, process data, including time, pH, temperature, dissolved oxygen, and agitation speed, were logged digitally. Final microbial counts were used to categorize the fermentation outcomes into four classes: (1) Successful: ≥3 × 10^9^ CFU/mL; (2) Semi-successful: 1–3 × 10^9^ CFU/mL; (3) Unsuccessful: <1 × 10^9^ CFU/mL; and (4) Complete failure: <1 × 10^7^ CFU/mL or undetectable. The classification was based on CFU counts after 48 h fermentation. The final dataset included 52 experiments (40 newly conducted and 12 from prior studies), with 38 used for training and 9 for testing. The complete failure group was excluded from training. The total of 52 fermentation runs was not derived from a formal design of experiments but from an empirical collection of available batch data under consistent operating parameters. Among them, 40 new runs were performed to encompass representative variations in early-stage process conditions (e.g., pH, temperature, dissolved oxygen), while 12 additional datasets from prior studies were incorporated to enhance diversity and model generalization. This approach reflects practical process variability rather than factorial-level combinations, ensuring that the CNN model learned from biologically and operationally realistic conditions.

### 2.4. Data Preprocessing and Feature Selection

Raw time-series data were preprocessed by removing outliers and imputing missing values using a linear interpolation approach based on adjacent time points. Outliers were identified when values exceeded three standard deviations from the mean. Each sample was truncated to the first 24 h (288 time points) for model input. Signal-to-Noise (S/N) ratio analysis was applied to identify features with the highest discriminatory power between outcome classes [18]. Features with minimal variation or constant values across all experiments (e.g., fixed setpoints) were removed. The final input dataset consisted of three key features: pH, actual temperature, and time. Outliers were identified as observations exceeding ±3 SD from the batch-specific mean and flagged prior to downstream processing (Figure 2b). Missing values were imputed using linear interpolation on the time axis, ensuring continuity of local signal dynamics (Figure 2c). To enable early prediction, signals were truncated to the first 24 h of fermentation (288 points at 5 min sampling; Figure 2d), corresponding to the model’s early decision window. Finally, each truncated sequence was normalized to the range [0–1] to standardize the scale across variables (Figure 2e).

#### Signal-to-Noise (S/N) Ratio for Feature Ranking

The S/N ratio was adopted as a feature-ranking metric because it provides a robust, model-independent measure of discriminative power under small-sample conditions. This approach, originating from Taguchi’s design-of-experiments methodology, quantifies the balance between inter-class signal strength and intra-class noise, thereby identifying variables that contribute most to class separation [19,20]. Compared with mutual information or recursive feature elimination (RFE), the S/N ratio avoids estimator bias and performs reliably even with limited experimental data. Specifically, mutual information can be sensitive to binning effects and small-sample variability, whereas RFE depends on model choice and parameter tuning, potentially introducing circularity in feature evaluation. In contrast, the S/N ratio offers computational simplicity, interpretability, and robustness in biological datasets characterized by stochastic fluctuations. In this study, the S/N ratio–derived feature order (pH, actual temperature, time) was consistent with that obtained from mutual information analysis (Spearman correlation = 0.91), supporting its validity as an efficient ranking criterion for fermentation datasets.

To evaluate the contribution of each recorded variable toward classification performance, we applied an S/N ratio method. This approach is commonly used in experimental design and feature selection to assess the discriminative capacity of input features. The S/N ratio for each variable was calculated using the following Equation:$$\frac{S}{N}=\;-10\cdot log10\;⎛\left(\frac{1}{n}\right)\Sigma_{\left\{i=1\right\}}^{\left\{n\right\}\left(1-y_{i}^{2}\right)}⎞$$
where *y_i_* denotes the response value (e.g., CFU concentration or target classification label) for the *i^th^* observation, and *n* is the total number of observations. This formulation, based on the “larger-the-better” criterion from Taguchi methods, emphasizes higher response values and penalizes variability.

After calculating the S/N ratios for all recorded variables, the top three features, pH value, actual fermentation temperature, and time, were selected based on their ranking scores. These variables exhibited the highest signal contributions and were retained as inputs for model training. Static parameters (e.g., setpoint temperature, RPM) and redundant signals were excluded from subsequent analysis. Figure 3 shows representative early-stage pH trajectories, which display informative dynamics and were consistently identified as among the most predictive variables by the S/N ratio method. Temperature and time followed similar monotonic trends in their S/N profiles, further supporting their selection as predictive features.

### 2.5. CNN Model Architecture and Training

A Convolutional Neural Network (CNN) model was constructed using one-dimensional time-series input vectors. The network architecture included: (1) Input layer: 3 × 288 features; (2) Two convolutional layers (kernel size = 3, filters = 32 and 64, activation = ReLU); (3) Max pooling layers (pool size = 2); (4) Flattening layer; (5) two fully connected dense layers, each consisting of 64 units with ReLU activation; and (6) Output layer (softmax, 3 classes). This dual-layer dense configuration was chosen to enhance feature abstraction and improve classification stability. The model was trained for 500 epochs using categorical cross-entropy loss and the Adam optimizer (learning rate = 0.001), with 80% of the dataset for training and 20% for validation. Training and implementation were carried out using Python (v3.9), Pandas (v2.2.3), NumPy(v1.23.0), SciPy (v1.16.2), Matplotlib (v3.3.2), TensorFlow/Keras (v2.12), and scikit-learn (v2.12). As depicted in Figure 4, the architecture integrates convolution, pooling, and dense layers, enabling efficient extraction of temporal features for early prediction of fermentation outcomes.

### 2.6. Model Validation and Intervention Protocol

To evaluate the real-time predictive capability of the model, two-stage validation was performed. In the first stage, the model classified fermentation outcomes based on data from the first 24 h. In the second stage, if a “semi-successful” or “failure” class was predicted, a manual intervention was carried out between 16 and 20 h, corresponding to the model’s earliest stable decision window. Corrective actions included (i) pH adjustment by aseptically adding 1 M NaOH in small aliquots to raise the medium from approximately pH 4.0–4.2 to 5.0 ± 0.1, and (ii) nutrient supplementation by introducing 1% (*v*/*v*) sterile-filtered MRS medium. Each fermentation received at most one corrective action. Comparative experiments were conducted with and without intervention across trials performed by novice and experienced operators. Improvement in microbial yield was assessed to validate the effectiveness of early prediction-based control.

For external validation, four additional datasets were collected from independent fermentation batches conducted at separate times within the same pilot-scale facility. Each dataset comprised 10–12 runs performed by different operators using fresh MRS medium preparations and new inoculum cultures. Slight variations in environmental and process conditions were intentionally retained—specifically, ambient temperature (±1.5 °C), agitation rate (180–220 rpm), and initial pH (6.3–6.7)—to evaluate model robustness under realistic operational fluctuations. None of these data were used for model training or hyperparameter tuning.

## 3. Results

### 3.1. Dataset Composition and Labeling

A total of 52 fermentation experiments were conducted, including 40 newly performed and 12 derived from prior published studies. Based on final CFU counts, fermentation outcomes were classified into four categories: successful (≥3 × 10^9^ CFU/mL), semi-successful (1–3 × 10^9^ CFU/mL), unsuccessful (<1 × 10^9^ CFU/mL), and complete failure (<1 × 10^7^ CFU/mL or undetectable). After excluding five complete failure cases, 47 datasets were retained. Of these, 38 were used for model training and 9 for testing.

### 3.2. Fermentation Dynamics (CFU over Time)

Viable cell counts of *L. plantarum* demonstrated a characteristic temporal pattern over the course of fermentation (Figure 5). During the fermentation process, the viable counts of *L. plantarum* increased from approximately 9.4 Log CFU/mL at 24 h to 9.5 Log CFU/mL at 48 h, followed by a decline to 9.0 Log CFU/mL at 72 h (Figure 5a). In addition to temporal dynamics, incubation temperature exerted a marked influence on cell viability, with higher counts observed at 37 °C compared to 30 and 32 °C (Figure 5b). These results highlight both time- and temperature-dependent factors shaping fermentation performance. This peak at mid-fermentation is consistent with nutrient utilization and acid accumulation dynamics typical of lactic acid bacteria. The observation supports the use of early (≤24 h) time-series signals for reliable prediction of final outcomes, which motivates our CNN-based early-stage modeling in subsequent sections.

### 3.3. Feature Selection and Signal Contribution

Among all recorded variables (e.g., pH, temperature, dissolved oxygen), three features, pH value, actual fermentation temperature, and fermentation time, demonstrated the highest discriminative power, based on signal-to-noise ratio analysis. These features were retained as model inputs, while redundant or static variables (e.g., setpoint values) were excluded.

### 3.4. CNN Model Performance

Using the first 24 h inputs (288 time points) of the three selected variables, the CNN (two convolutional layers + max pooling + dense + softmax) achieved a test accuracy of 97.87% (Figure 6), exceeding all baseline models (LSTM, 93.61%; XGBoost, 95.74%; LightGBM, 93.61%). Figure 6 shows the side-by-side comparison of model accuracies on the internal test set, clearly highlighting the advantage of the CNN for early prediction.

### 3.5. ROC-Based Discrimination

Receiver-Operating-Characteristic analysis indicated strong discrimination across classes: AUC = 0.99 (failure), 0.98 (semi-successful), and 1.00 (successful); macro-average AUC = 0.98, micro-average AUC = 0.99. Figure 7 shows the ROC curves for each class together with macro/micro averages, corroborating the CNN’s sensitivity-specificity balance.

### 3.6. Classification Error Structure

To complement AUC-level performance, Figure 8 illustrates the confusion matrix of the CNN on the internal evaluation set (*n* = 47), which included all fermentation runs not used in model training. The results show that most samples were correctly classified along the diagonal: 4 failures (class 0), 26 semi-successful (class 1), and 14 successful fermentations (class 2). Only three misclassifications were observed, comprising one semi-successful run predicted as failure, one successful run predicted as failure, and one successful run predicted as semi-successful.

This distribution indicates that errors were limited and occurred only between adjacent outcome categories. Importantly, no gross misclassifications were observed (e.g., failures being classified directly as successes), underscoring the robustness of the CNN in capturing meaningful fermentation patterns. Figure 8 thus highlights the practical reliability of the model by demonstrating both high accuracy and biologically plausible error modes.

The CNN model demonstrated strong generalization when applied to four independent validation datasets collected under slightly varied process conditions, achieving an average accuracy of 96.8 ± 1.2%, consistent with internal cross-validation results.

### 3.7. Learned Temporal Patterns (Qualitative Visualization)

To provide qualitative insight into model behavior, Figure 9 presents a schematic visualization of representative temporal motifs that the CNN is expected to exploit within the first 24 h. These motifs summarize recurring local oscillations and trend shifts observed in the data and align with the network’s convolutional inductive bias. These patterns suggest that discriminative cues emerge early and are repeatedly exploited by the network to separate marginal or failing trajectories from successful ones. These qualitative motifs explain why separability emerges early and recurs across trajectories, supporting the feasibility of early-stage decision-making.

### 3.8. Impact of Intervention Based on Model Predictions

As shown in Figure 10, CNN-guided interventions achieved significantly higher final *L. plantarum* CFU/mL compared to control runs across both operator groups. For novice operators, mean yields increased from 5.57 × 10^8^ to 1.75 × 10^9^ CFU/mL (*p* < 0.05), while for expert operators, yields rose from 1.31 × 10^9^ to 3.39 × 10^9^ CFU/mL (*p* < 0.01). These improvements underscore that the intervention effect was consistent regardless of operator experience, highlighting the robustness and practical utility of the proposed AI-driven strategy.

## 4. Discussion

This study demonstrates that a one-dimensional CNN trained on the first 24 h of fermentation time-series (pH, temperature, time; 288 points) can accurately classify L. plantarum LP198 outcomes into three categories and outperforms LSTM, XGBoost, and LightGBM under identical data splits. On the internal test set, the CNN achieved 97.87% accuracy, compared with 93.61% (LSTM), 95.74% (XGBoost), and 93.61% (LightGBM) (Figure 6). Discrimination was consistently high across classes, with AUC = 0.99 (failure), 0.98 (semi-successful), and ≈1.00 (successful); the macro-average AUC = 0.98 and micro-average AUC = 0.99 (Figure 7). The confusion matrix shows only limited misclassification—primarily between adjacent biological categories—consistent with expected gradations near decision boundaries (Figure 8). These misclassifications occurred only between adjacent categories, consistent with biological variability at borderline outcomes, and did not involve gross errors (e.g., failure classified as success). Together, these results indicate that discriminative cues emerge within the first 24 h, supporting the use of CNNs as soft sensors for early decision-making in probiotic fermentation [8,9,17]. Although the total number of fermentation runs (*n* = 47 effective) may appear limited, this scale aligns with prior AI-driven fermentation studies, where each experiment involves complex bioreactor operations and biological variability [15,16]. The model’s consistent performance across independent training–testing splits and fivefold cross-validation suggests that the CNN learned generalizable temporal patterns rather than overfitting to batch-specific noise. In addition, regularization measures such as dropout and early stopping were applied to enhance robustness. Nevertheless, broader validation with expanded datasets across strains and facilities will be pursued in future work to further strengthen generalization. Furthermore, the CFU trajectory across 24–72 h (Figure 5a) corroborates the biological basis for early prediction: viability typically peaks by ~48 h and declines thereafter, indicating that discriminative cues emerge well before endpoint measurements. The observed CFU trends (Figure 5) are consistent with known growth optima of *L. plantarum*, where mid-fermentation (48 h) and moderate-to-high incubation temperatures (37 °C) yield the most favorable outcomes. This biological behavior underscores the relevance of incorporating temperature and early growth trajectories into predictive models.

Compared with previously reported ML- and DL-based fermentation studies [15,16,19], the present CNN framework offers distinctive advantages in both functionality and applicability. It enables early-stage prediction (within the first 24 h) using minimal sensor data, directly captures temporal trajectory features instead of static endpoints, and allows real-time corrective intervention based on model guidance. Moreover, its validation across four independent datasets confirms robustness and generalizability, supporting its potential use as a practical control tool for industrial-scale probiotic fermentation.

Relative to traditional modeling approaches widely used in fermentation optimization—such as ANN with RSM/GA hybrids and classical response-surface–based designs—the present framework addresses two practical limitations reported in prior work: reliance on endpoint readouts and incomplete treatment of temporal dependencies [3,14,16,19,20]. Prior ANN/RSM studies have successfully optimized media or targeted metabolite yields in lactic acid bacteria and related systems, yet they typically treat inputs as static factors and therefore do not leverage early process trajectories [14,16,19,20]. By contrast, the CNN’s convolutional filters capture localized temporal motifs in sensor streams (Figure 9), providing a mechanistic–agnostic but trajectory-aware representation that aligns with recent observations of CNN efficacy on biological time-series and lactic acid bacteria processes [8,9]. In this context, the present findings extend the ANN-centric literature by showing that temporal feature learning can deliver deployable, early-stage classification with industrial relevance [3,8,9].

Importantly, model-guided interventions translated algorithmic gains into operational improvements. When early predictions triggered simple corrective actions (e.g., pH adjustment, nutrient supplementation), final CFU/mL increased for both novice and expert operators, 5.57 × 10^8^ → 1.75 × 10^9^ (novice) and 1.31 × 10^9^ → 3.39 × 10^9^ (expert), demonstrating that decision support can reduce operator-dependent variability and elevate yield (Figure 10). This effect, observed under routine conditions, suggests a practical route to standardizing probiotic manufacturing, complementing the optimization-oriented strategies reported previously for LAB processes [14,16,19,20]. From a quality-by-design perspective, such soft-sensing closes the loop between early diagnostics and targeted control actions, thereby lowering the risk of batch failure while improving resource efficiency [8,9].

External validation with four independent datasets further supports generalizability, with correct outcome predictions at early checkpoints (e.g., ~40th time point) and at 24 h. While the external sets differ in strain/context, the model maintained high agreement with final CFU-based categories, indicating robustness of the learned temporal signatures rather than overfitting to internal idiosyncrasies. This observation is consistent with reports that deep models, properly regularized, can transfer across related fermentation regimes when key process trajectories are retained [8,9,17].

Several limitations warrant consideration. First, the internal dataset comprises 47 valid runs for LP198; although sufficient to demonstrate utility, broader coverage across strains, substrates, and scales is required to fully characterize generalization boundaries. Second, only three sensor variables were used; incorporating dissolved oxygen, redox, on-line spectroscopy, or metabolomics could further improve sensitivity to failure precursors [3,8]. Third, while the current CNN effectively captures local patterns, attention-based or hybrid CNN-Transformer architectures may better model long-range dependencies and cross-channel interactions, as suggested by recent deep learning advances [17]. Finally, baseline implementations (LSTM, XGBoost, LightGBM) followed standard practice [18], but expanded hyperparameter sweeps and probabilistic calibration could sharpen comparative insights, especially for borderline “semi-successful” cases (Figure 8). We also note that Figure 9 is a qualitative schematic rather than raw intermediate activations; activation-based interpretability (e.g., layer-wise relevance, integrated gradients) will be reported in follow-up work using released weights and code.

In summary, the proposed CNN enables early, accurate, and actionable prediction of probiotic fermentation outcomes using only the first 24 h of process data, outperforming strong baselines (Figure 5, Figure 6 and Figure 7) and improving yield via model-guided interventions (Figure 9). These findings build on and extend prior ANN/RSM-based optimization studies by introducing a trajectory-aware soft-sensing paradigm suited for real-time manufacturing control in *L. plantarum* and, potentially, other industrial microbes [3,8,9,14,16,19,20]. Practical deployment of the CNN model can be achieved by integrating it into existing fermentation control software as an early-warning and decision-support module. The system continuously receives sensor inputs (pH, temperature, and dissolved oxygen) and provides classification outputs within the first 24 h of fermentation. When suboptimal outcomes are predicted, operators can apply targeted corrective actions, such as pH correction, nutrient supplementation, or temperature adjustment, based on the system’s alerts. This workflow can also be embedded into Supervisory Control and Data Acquisition (SCADA) or Programmable Logic Controller (PLC) platforms to enable semi-automated or fully automated control, thereby improving process robust-ness and consistency in industrial probiotic production. Future work will scale datasets across strains and facilities, integrate richer sensing modalities, and explore attention-based architectures to further enhance early-stage reliability and deployment readiness [17,18].

## 5. Conclusions

This study demonstrates that a one-dimensional convolutional neural network (CNN) trained on the first 24 h of fermentation time-series (pH, temperature, time; 288 points) can accurately classify *L. plantarum* LP198 outcomes into successful, semi-successful, and unsuccessful categories. Under identical data splits, the CNN outperformed strong baselines, LSTM, XGBoost, and LightGBM, on the internal test set, achieving 97.87% accuracy versus 93.61%, 95.74%, and 93.61%, respectively. ROC analysis further indicated high discrimination across classes (AUC = 0.99 for failure, 0.98 for semi-successful, and ≈1.00 for successful), with macro-average AUC = 0.98 and micro-average AUC = 0.99. These results support the feasibility of early, trajectory-aware soft sensing for probiotic fermentation using routine process signals.

Importantly, early predictions translated to measurable operational gains when used to guide simple interventions (e.g., pH adjustment, nutrient supplementation). Final CFU/mL increased from 5.57 × 10^8^ (control) to 1.75 × 10^9^ (intervention) in novice-operated runs, and from 1.31 × 10^9^ to 3.39 × 10^9^ in expert-operated runs. These improvements indicate that model-guided decision support can reduce operator-dependent variability and enhance yield without altering core process hardware or adding invasive sensors.

This work has practical scope and limitations. The modeling was performed on 47 valid runs drawn from 52 experiments of a single strain and used three on-line variables; broader validation across strains, substrates, and facilities, and the inclusion of additional sensing (e.g., DO/redox, spectroscopic or metabolomic signals) are warranted. Future efforts will expand dataset scale and diversity, evaluate prospective deployments, and explore attention-based or hybrid CNN-Transformer architectures to capture longer-range dependencies while maintaining real-time applicability. Overall, the present framework provides a deployable pathway for early prediction and targeted control in industrial probiotic fermentation.

## Figures and Tables

**Figure 1 microorganisms-13-02601-f001:**
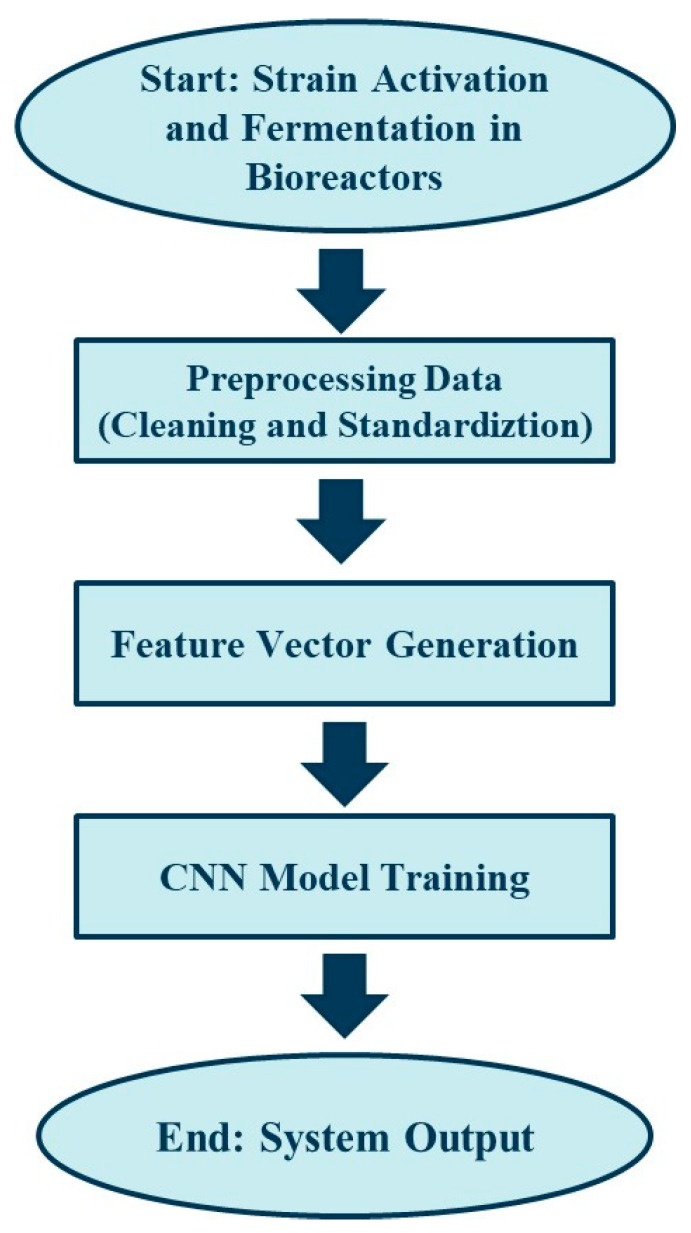
Workflow of the probiotic fermentation monitoring and deep learning prediction system. The process begins with fermentation experiments and data collection, followed by preprocessing and feature vector generation. A CNN-based framework is then trained to provide predictive outputs, enabling real-time monitoring and optimization.

**Figure 2 microorganisms-13-02601-f002:**
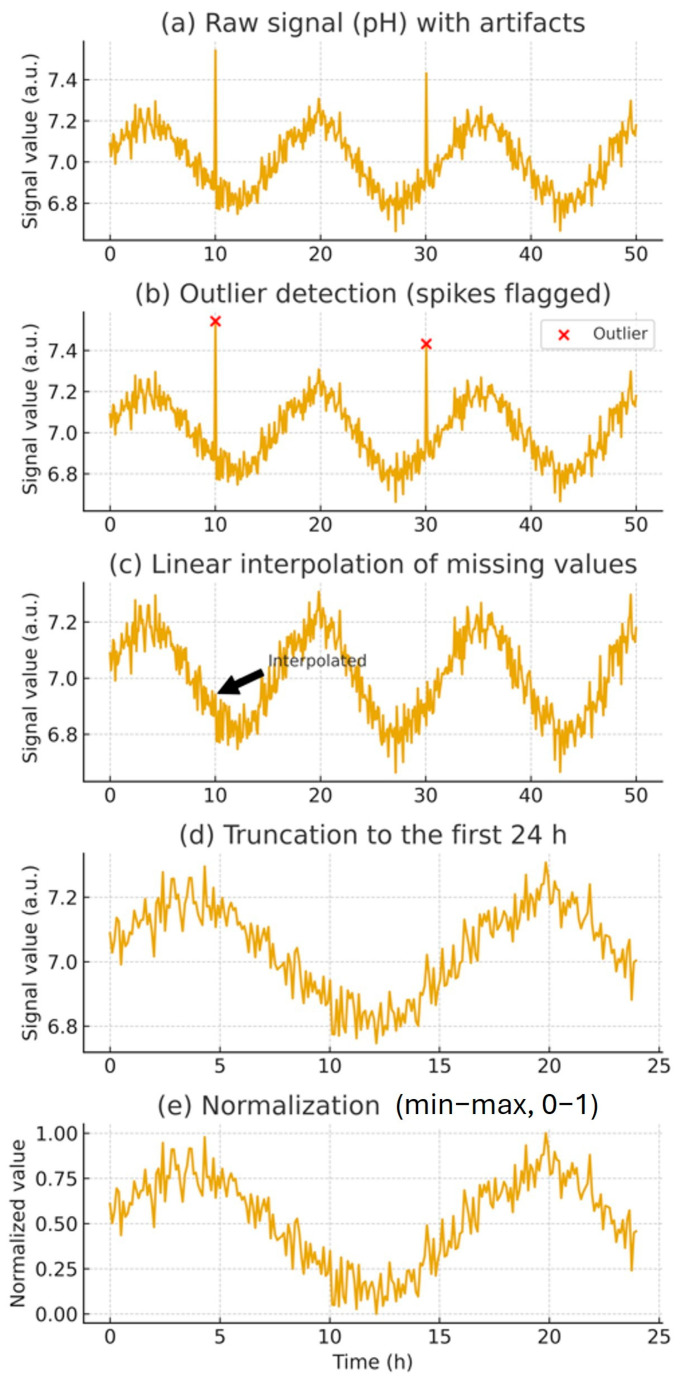
Preprocessing pipeline of fermentation time-series data. (**a**) Raw signal with noise and artifacts. (**b**) Outlier detection, with spikes flagged (red ×). (**c**) Linear interpolation of missing values, with arrows marking corrected points. (**d**) Truncation to the first 24 h of data. (**e**) Normalization of truncated signals using min–max scaling to the [0–1] range.

**Figure 3 microorganisms-13-02601-f003:**
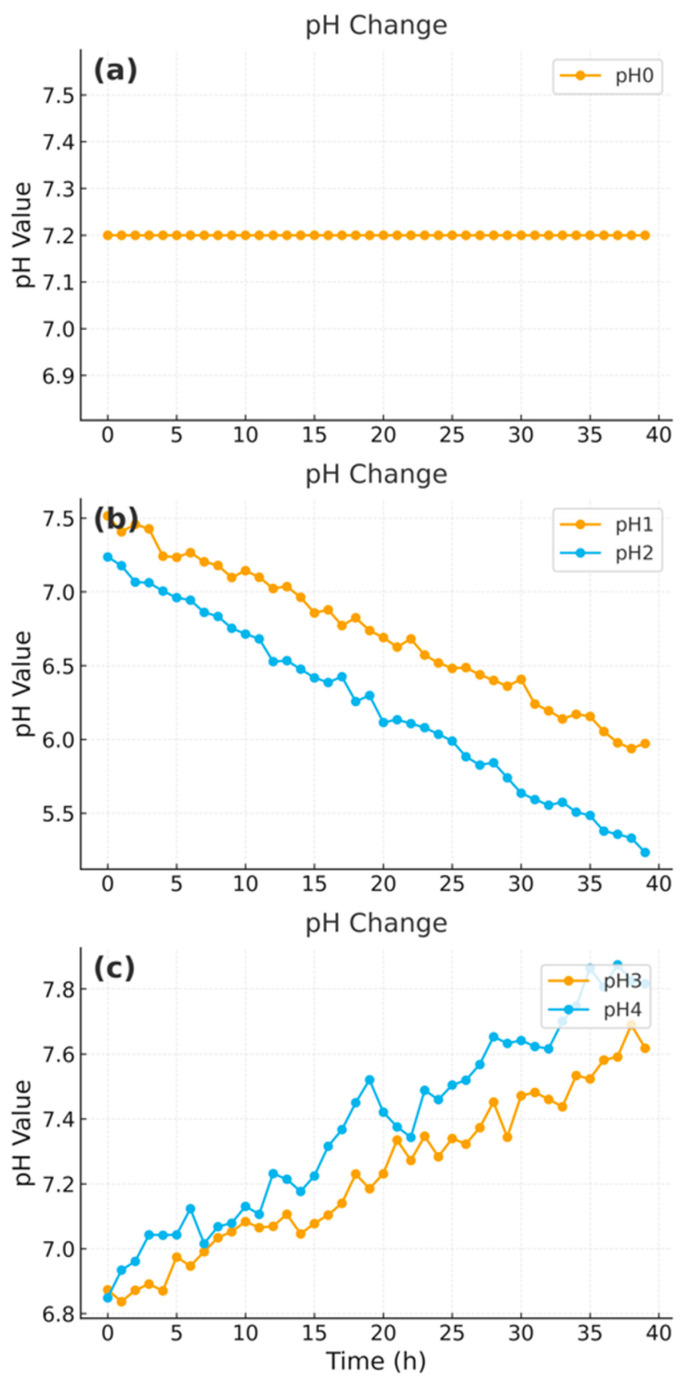
Representative early-stage pH trajectories across fermentation runs: (**a**) pH0, (**b**) pH1–pH2, (**c**) pH3–pH4.

**Figure 4 microorganisms-13-02601-f004:**
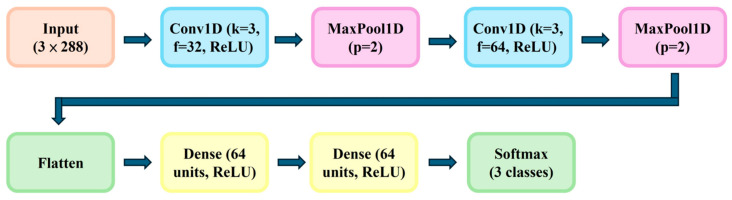
CNN architecture for early prediction of fermentation outcomes, consisting of two convolutional layers (kernel size = 3; filters = 32 and 64; ReLU), each followed by MaxPool1D (pool size = 2), a flatten operation, two dense layers (each with 64 units, ReLU), and a final softmax classifier for three classes. The model ingests early-stage features of dimension 3 × 288.

**Figure 5 microorganisms-13-02601-f005:**
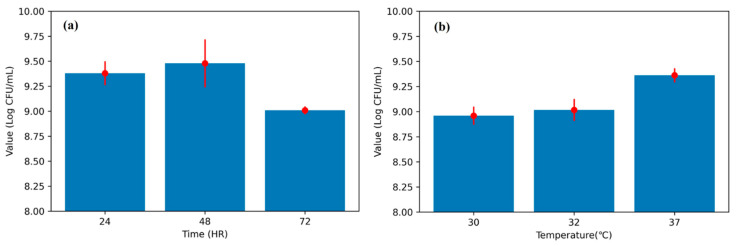
Fermentation dynamics of *L. plantarum.* (**a**) Changes in viable cell counts over time (24, 48, and 72 h). (**b**) Effect of incubation temperature (30, 32, and 37 °C) on final cell counts. Error bars indicate the mean ± standard deviation.

**Figure 6 microorganisms-13-02601-f006:**
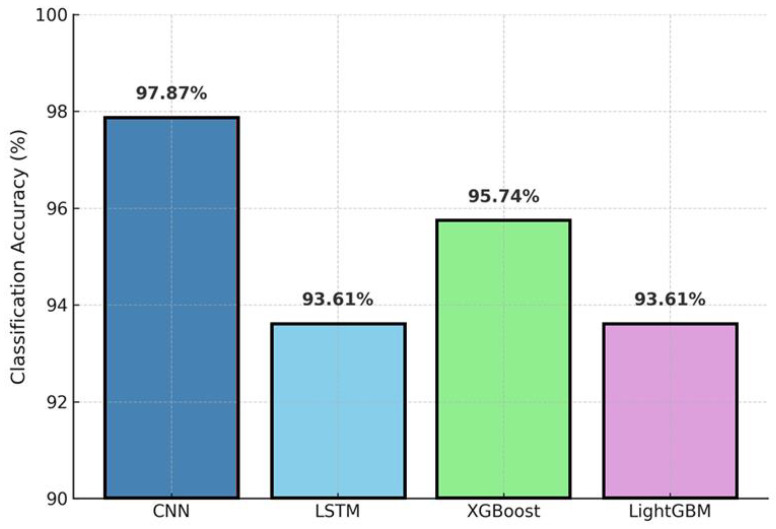
Classification accuracy comparison among CNN, LSTM, XGBoost, and LightGBM on the internal test set using the first 24 h inputs. The CNN achieved 97.87%, outperforming all baselines.

**Figure 7 microorganisms-13-02601-f007:**
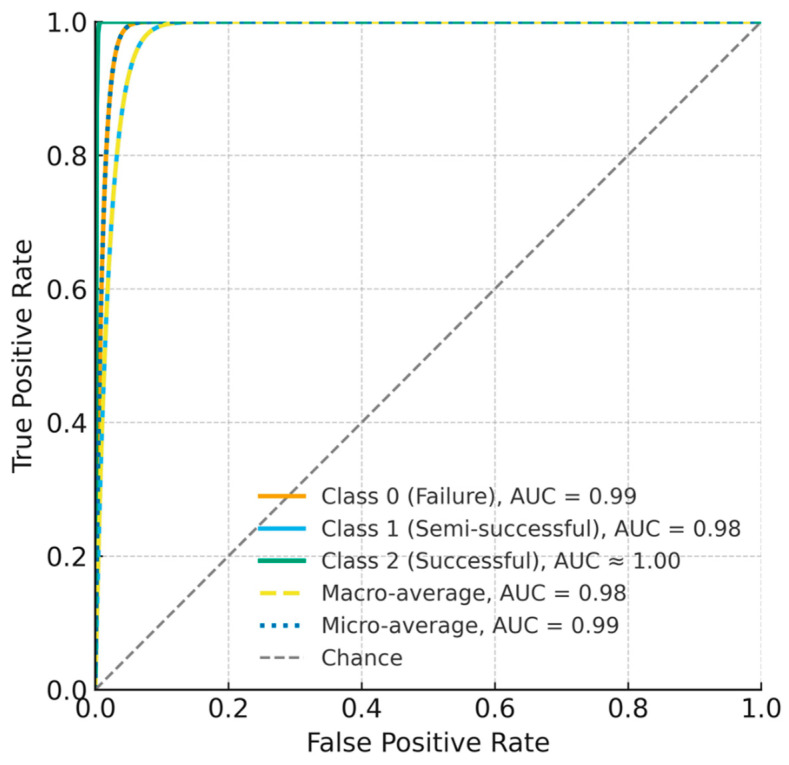
ROC curves for the three outcome classes, failure (class 0, orange solid line), semi-success (class 1, blue solid line), and success (class 2, green solid line), based on CNN predictions on the test set. The macro-average ROC (yellow dashed line) and micro-average ROC (blue dotted line) yielded AUC values of 0.98 and 0.99, respectively, while the grey dashed line indicates random-chance performance (AUC = 0.50).

**Figure 8 microorganisms-13-02601-f008:**
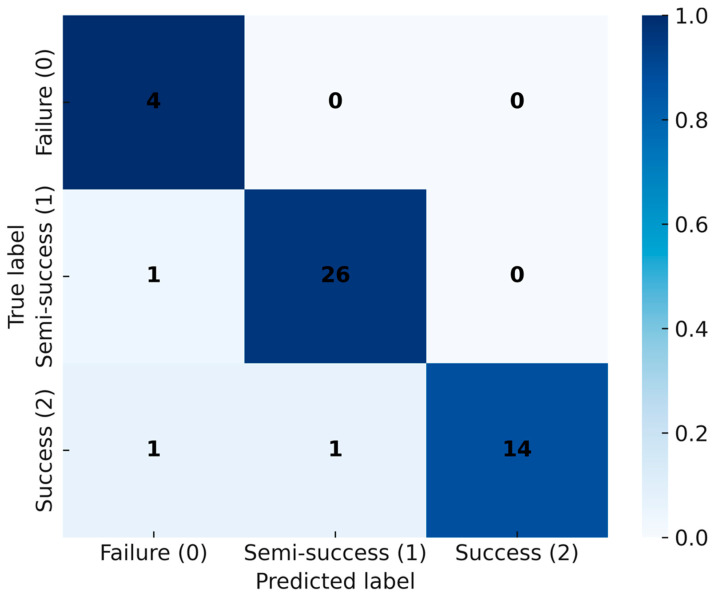
Confusion matrix of CNN classification results on the internal evaluation set (*n* = 47). Cell values are counts, while the color scale indicates row-normalized proportions (0–1). Classes correspond to fermentation outcomes: class 0 = failure, class 1 = semi-success (semi-successful), and class 2 = success.

**Figure 9 microorganisms-13-02601-f009:**
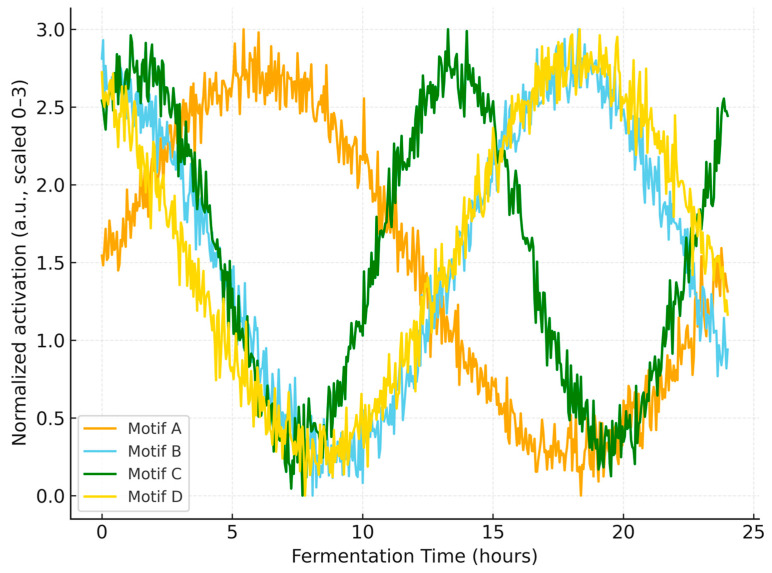
Representative temporal motifs extracted by the CNN from the first 24 h of fermentation (288 time points). The four curves (Motifs A–D) illustrate normalized activation patterns (scaled to 0–3 a.u.) that the network used for early discrimination of fermentation outcomes. These temporal features highlight recurring oscillatory and trending signals within the early phase, supporting the model’s ability to achieve accurate predictions prior to the completion of the fermentation process.

**Figure 10 microorganisms-13-02601-f010:**
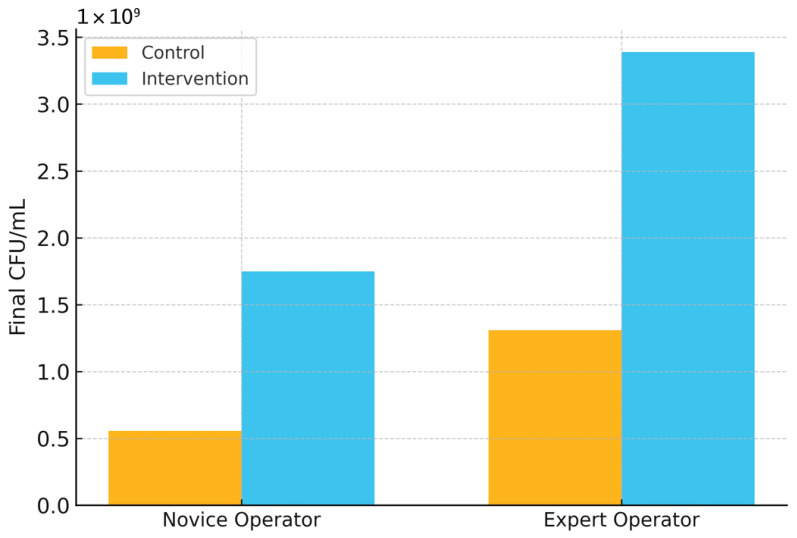
Impact of intervention based on model prediction across operator groups. Final *L. plantarum* CFU/mL in control versus intervention runs for novice and expert operators. Axis uses scientific notation (×10^9^). Values correspond to group means reported in 3.8. (novice: 0.56 → 1.75; expert: 1.31 → 3.39 × 10^9^ CFU/mL).

## Data Availability

The original contributions presented in this study are included in the article. Further inquiries can be directed to the corresponding author.

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
