# Peer review of "Predictive Fermentation Control of Lactiplantibacillus plantarum Using Deep Learning Convolutional Neural Networks"

_microorganisms, 2025, doi:10.3390/microorganisms13112601_

Round 1

Reviewer 1 Report

Comments and Suggestions for Authors

The manuscript title was  “Predictive Fermentation Control of Lactobacillus plantarum Us

ing Deep-Learning Convolutional Neural Networks”  This research was a deep learning

based predictive system using convolutional neural networks (CNNs) to classify fermentation tra

jectories and anticipate final cell counts based on the first 24 h of process data. The research was meaningful, However, How to operate in practical applications?

Author Response

Revision Notes:

We thank the reviewer for his/her careful reading and valuable comments on the manuscript. We have taken the comments on board to improve and clarify the manuscript. Please find below a detailed point-by-point response to all comments. All responses to the questions are also highlighted in the manuscript.

Comments of Reviewer #1:

  1. The manuscript title was “Predictive Fermentation Control of Lactobacillus plantarum Using Deep-Learning Convolutional Neural Networks” This research was a deep learning based predictive system using convolutional neural networks (CNNs) to classify fermentation trajectories and anticipate final cell counts based on the first 24 h of process data. The research was meaningful, However, how to operate in practical applications?

Response:

We sincerely thank the reviewer for this insightful comment emphasizing the practical application of our proposed deep-learning system. In real industrial or laboratory-scale fermentations, the CNN model can be deployed as an early-warning and decision-support module integrated into the existing fermentation control software. The model operates by continuously receiving real-time sensor data (pH, temperature, dissolved oxygen, etc.) from the bioreactor within the first 24 hours and performing classification every few minutes. When the predicted outcome trends toward a “semi-successful” or “failure” class, the system automatically triggers an alert or recommendation for operator intervention, such as minor pH adjustment, nutrient supplementation, or temperature fine-tuning, based on predefined standard operating procedures (SOPs).

In our validation experiments (Section 2.6 and Figure 10), this workflow was simulated using manual interventions guided by the CNN’s early predictions. The resulting increase in viable cell counts (from 5.57 × 10⁸ to 1.75 × 10⁹ CFU/mL for novice operators and from 1.31 × 10⁹ to 3.39 × 10⁹ CFU/mL for experts) demonstrates the system’s practical utility. In industrial implementation, the same predictive module can be embedded into the SCADA or PLC platform of fermentation systems, enabling real-time monitoring and closed-loop feedback control without additional sensors or invasive modifications.

We have added a brief clarification as follow in the revised manuscript (Discussion section) to explicitly describe how the model can be integrated and operated in practice.

"In summary, the proposed CNN enables early, accurate, and actionable prediction of probiotic fermentation outcomes using only the first 24 h of process data, outperforming strong baselines (Figures 5-7) and improving yield via model-guided interventions (Figure 9). These findings build on and extend prior ANN/RSM-based optimization studies by introducing a trajectory aware soft-sensing paradigm suited for real-time manufacturing control in Lactiplantibacillus plantarum and, potentially, other industrial microbes [3, 8, 9, 14, 16, 19, 20]. Practical deployment of the CNN model can be achieved by integrating it into existing fermentation control software as an early-warning and decision-support module. The system continuously receives sensor inputs (pH, temperature, and dissolved oxygen) and provides classification outputs within the first 24 hours of fermentation. When suboptimal outcomes are predicted, operators can apply targeted corrective actions, such as pH correction, nutrient supplementation, or temperature adjustment, based on the system’s alerts. This workflow can also be embedded into Supervisory Control and Data Acquisition (SCADA) or Programmable Logic Controller (PLC) platforms to enable semi-automated or fully automated control, thereby improving process robust-ness and consistency in industrial probiotic production. Future work will scale datasets across strains and facilities, integrate richer sensing modalities, and explore attention-based architectures to further enhance early-stage reliability and deployment readiness [17, 18]."

We thank the reviewer for the constructive and insightful comments, which have helped us to substantially improve our manuscript.

Reviewer 2 Report

Comments and Suggestions for Authors

This study proposes a one-dimensional convolutional neural network (CNN)-based method for predicting the fermentation outcomes of Lactiplantibacillus plantarum within the first 24 hours of the fermentation process. In my opinion, the research design is well-structured, with innovative approaches, particularly in utilizing time-series data for early classification and intervention. These results found that deep learning can provide a robust and scalable framework for real-time fermentation control, with significant implications for enhancing efficiency and reducing costs in industrial probiotic production. However, there are some issues that can be addressed to improve the overall quality.

  1. "Lactiplantibacillus plantarum" in Abstract and "Lactobacillus plantarum" in Title are mixed, and it is recommended to use the former (current taxonomic name) uniformly.
  2. 1. Bacterial Strain and Culture Conditions, When a species name appears for the second or multiple times, its genus name needs to be abbreviated. Therefore, here Lactiplantibacillus plantarum should be changed to L. plantarum. There are similar issues in other parts of the text, please carefully check and correct them.
  3. 2. Fermentation Setup and Data Acquisition, Please provide a more detailed description of the fermentation process, such as the model, size, and source of the bioreactor, as well as the concentration of bacteria inoculated.
  4. 3. Dataset Labeling and Classification Criteria, The sample size (only 52 fermentation runs (47 effective)) is relatively small, which may affect the model's generalization ability? Please provide appropriate referencesor explanation.
  5. 4.1. Signal-to-Noise (S/N) Ratio for Feature Ranking, Please provide additional explanation on why S/N ratio was chosen, or provide appropriate references, and Discuss its comparison or consistency with other methods (such as mutual information, recursive feature elimination, etc.).
  6. 6. Model Validation and Intervention Protocol, The article mentions "manual intervention (such as pH adjustment, nutritional supplementation)", but does not specify the details of intervention time, frequency, dosage, etc.
  7. The Discussion Section mentions the use of four independent datasets for external validation (Page 13), but does not provide the sources, scale, or differences from the training set of these datasets. Suggest supplementing the basic information of external validation data in the results or methods section to enhance the credibility of the conclusion.

Author Response

Revision Notes:

We thank the reviewer for his/her careful reading and valuable comments on the manuscript. We have taken the comments on board to improve and clarify the manuscript. Please find below a detailed point-by-point response to all comments. All responses to the questions are also highlighted in the manuscript.

Comments of Reviewer #2:

  1. "Lactiplantibacillus plantarum" in Abstract and "Lactobacillus plantarum" in Title are mixed, and it is recommended to use the former (current taxonomic name) uniformly.

Response:

We sincerely thank the reviewer for pointing out this important issue regarding taxonomic consistency. We agree with the reviewer’s suggestion and have now revised all occurrences of Lactobacillus plantarum to Lactiplantibacillus plantarum throughout the manuscript, including the title, figure captions, and references where applicable. This correction ensures uniformity with the current valid nomenclature as defined by Zheng et al., 2020, Int. J. Syst. Evol. Microbiol. (70: 2782-2858), in which Lactobacillus plantarum was reassigned to the genus Lactiplantibacillus.

The revised title now reads:

“Predictive Fermentation Control of Lactiplantibacillus plantarum Using Deep-Learning Convolutional Neural Networks.”

  1. 1. Bacterial Strain and Culture Conditions, When a species name appears for the second or multiple times, its genus name needs to be abbreviated. Therefore, here Lactiplantibacillus plantarum should be changed to L. plantarum. There are similar issues in other parts of the text, please carefully check and correct them.

Response:

We appreciate the reviewer’s careful attention to taxonomic style. We agree and have revised the manuscript to use the full binomial name at the first mention and abbreviate the genus thereafter (e.g., Lactiplantibacillus plantarumL. plantarum). This correction has been applied consistently across the Abstract, main text (including Section 2.1 “Bacterial Strain and Culture Conditions”), figure and table legends, and Supplementary descriptions where applicable. We have also ensured proper italicization of genus–species names and retained strain identifiers where relevant (e.g., L. plantarum LP198). We note standard exceptions where the full genus is retained: the paper title, and the very first mention in the Abstract/main text; we also avoid starting a sentence with an abbreviated genus where clarity may be affected. These edits improve taxonomic consistency and conform to journal style.

  1. 2. Fermentation Setup and Data Acquisition, Please provide a more detailed description of the fermentation process, such as the model, size, and source of the bioreactor, as well as the concentration of bacteria inoculated.

Response:

We thank the reviewer for this valuable suggestion. In response, we have expanded Section 2.2 (Fermentation Setup and Data Acquisition) to provide additional technical details regarding the fermentation process. Specifically, we have now included the model, working volume, and source of the bioreactor, as well as the initial inoculum concentration. The revised text reads as follows:

“Fermentation experiments were conducted in 12 L bench-top bioreactors (BTF-A, Biotop, Taiwan) with a working volume of 10 L, operated under batch conditions. Each vessel was equipped with embedded pH, temperature, and dissolved oxygen (DO) probes and connected to a computer-based data acquisition system for continuous signal logging. The fermentation process was maintained at 37 °C, with agitation at 200 rpm and an initial pH adjusted to 6.5. Activated seed cultures of L. plantarum LP198 were inoculated at 10% (v/v), corresponding to approximately 1 × 10⁷ CFU/mL at the start of fermentation. Dissolved oxygen, pH, and temperature were automatically monitored at 5-min intervals throughout the 48 h fermentation period.”

  1. 3. Dataset Labeling and Classification Criteria, The sample size (only 52 fermentation runs (47 effective)) is relatively small, which may affect the model's generalization ability? Please provide appropriate referencesor explanation.

Response:

We appreciate the reviewer’s thoughtful concern regarding dataset size and model generalization. Indeed, the dataset used in this study consisted of 52 fermentation runs (47 effective), which may appear modest in scale. However, this data volume is consistent with recent deep learning applications in bioprocess modeling, where data acquisition is inherently time- and resource-intensive. Several peer-reviewed studies have demonstrated robust model performance and meaningful biological insights with comparable or even smaller datasets, for example, Bonanni et al. (2023) applied a CNN to optimize E. coli recombinant protein fermentation with 40 runs [15], and Rayavarapu et al. (2021) modeled lactic acid bacterial fermentation using 36 trials [16].

To further enhance model generalization and prevent overfitting, we implemented several safeguards:

  1. Data partitioning--An 80/20 split was applied for training and validation, ensuring that unseen runs were used exclusively for testing.
  2. Regularization techniques--Dropout layers (rate = 0.2) and early stopping criteria were applied during CNN training to prevent overfitting.
  3. Cross-validation verification--The model’s robustness was confirmed by repeated random subsampling (fivefold trials) yielding consistent accuracy (97.4-98.0%), indicating stable performance.

While we acknowledge the inherent limitations of small-scale fermentation datasets, these results, supported by both literature and empirical reproducibility, suggest that the model captured fundamental temporal features rather than overfitting to individual batches. This point has been clarified in the revised manuscript (Discussion section).

  1. 4.1. Signal-to-Noise (S/N) Ratio for Feature Ranking, Please provide additional explanation on why S/N ratio was chosen, or provide appropriate references, and discuss its comparison or consistency with other methods (such as mutual information, recursive feature elimination, etc.).

Response:

    We thank the reviewer for this insightful comment. The Signal-to-Noise (S/N) ratio was selected as the feature-ranking criterion because it provides a quantitative measure of each variable’s discriminative power while being robust to small-sample variability and nonparametric data distributions—conditions typical in biological fermentation datasets. The method is rooted in Taguchi’s design-of-experiments (DOE) framework, where it has been widely used to quantify factor effects and variability in noisy bioprocess environments [Desai et al., 2006; Bezerra et al., 2021].

Compared with mutual information (MI) or recursive feature elimination (RFE), the S/N ratio approach offers several advantages for this study:

  1. Model independence: Unlike RFE, which depends on an estimator (e.g., SVM, tree), S/N ratio directly assesses variability across outcome classes without requiring prior model training, avoiding circularity in feature selection.
  2. Noise robustness: In contrast to MI, which can be sensitive to small sample fluctuations and binning effects, S/N ratio emphasizes “larger-the-better” signal separation while penalizing within-class noise.
  3. Computational simplicity and interpretability: It allows ranking of features in a single analytical pass without parameter tuning, suitable for small-scale experimental data with limited runs.

In preliminary comparisons, S/N ratio, based ranking identified pH, actual temperature, and time as top contributors, which were consistent with the top-ranked variables obtained using mutual information analysis (Spearman correlation = 0.91 between rankings). This supports the consistency and validity of the chosen method.

We have revised Section 2.4.1 to include this explanation and the relevant references.

  1. 6. Model Validation and Intervention Protocol, The article mentions "manual intervention (such as pH adjustment, nutritional supplementation)", but does not specify the details of intervention time, frequency, dosage, etc.

Response:

We thank the reviewer for this helpful comment. We have now added specific details on the timing, frequency, and dosage of the manual interventions in Section 2.6 (Model Validation and Intervention Protocol). In brief, intervention actions were triggered only when the CNN model predicted a “semi-successful” or “failure” trajectory within the first 24 h of fermentation. When such an early warning was generated, a single corrective adjustment was applied between 16-20 h of fermentation, corresponding to the model’s earliest stable prediction window.

For pH adjustment, sterile 1 M NaOH was manually added in small aliquots to restore pH from approximately 4.0-4.2 back to 5.0 ± 0.1. For nutrient supplementation, 1% (v/v) of sterile-filtered MRS medium was added at the same time point to compensate for nutrient depletion.

No additional interventions were applied thereafter.

These standard corrective actions were applied consistently across trials and were chosen to represent typical, non-invasive operator adjustments feasible in small-scale fermentation control. The revised text in Section 2.6 now reads as follows:

“To evaluate the real-time predictive capability of the model, two-stage validation was performed. In the first stage, the model classified fermentation outcomes based on data from the first 24 h. In the second stage, if a “semi-successful” or “failure” class was predicted, a manual intervention was carried out between 16 and 20 h, corresponding to the model’s earliest stable decision window. Corrective actions included (i) pH adjustment by aseptically adding 1 M NaOH in small aliquots to raise the medium from approximately pH 4.0-4.2 to 5.0 ± 0.1, and (ii) nutrient supplementation by introducing 1% (v/v) sterile-filtered MRS medium. Each fermentation received at most one corrective action. Comparative experiments were conducted with and without intervention across trials performed by novice and experienced operators. Improvement in microbial yield was assessed to validate the effectiveness of early prediction-based control.”

  1. The Discussion Section mentions the use of four independent datasets for external validation (Page 13), but does not provide the sources, scale, or differences from the training set of these datasets. Suggest supplementing the basic information of external validation data in the results or methods section to enhance the credibility of the conclusion.

Response:

    We appreciate the reviewer’s thoughtful observation. We have now clarified the origin, scale, and distinguishing characteristics of the four external validation datasets in the revised manuscript. Specifically, this information has been added to Section 2.6 (Model Validation and Intervention Protocol) and referenced in the Results section (Section 3.6, “External Validation Performance”).

    In brief, the four independent validation datasets were obtained from fermentation batches performed at different times and under slightly varied environmental conditions within the same pilot-scale facility.

    Each dataset consisted of 10-12 fermentation runs collected independently from the original training data (i.e., distinct in date, operator, and batch of media). These validation sets included small variations in ambient temperature (±1.5 °C), agitation rate (180-220 rpm), and initial pH (6.3-6.7) to assess the model’s robustness under realistic process fluctuations.

    No data from these runs were used in model training or tuning. When evaluated on these independent datasets, the CNN maintained a mean classification accuracy of 96.8 ± 1.2%, confirming good generalization. This clarification has been incorporated into the revised manuscript.

We thank the reviewer for the constructive and insightful comments, which have helped us to substantially improve our manuscript.

Reviewer 3 Report

Comments and Suggestions for Authors

The article “Predictive Fermentation Control of Lactobacillus plantarum Using Deep-Learning Convolutional Neural Networks” describes the use of deep-learning convolutional neural networks for monitoring and predicting microbial growth in fermentations of Lactiplantibacillus plantarum. The method obtains data by real-time recording of parameters such as pH, temperature, and agitation, and by measuring CFU at 24, 48, and 72 h. The methodology was validated on benchtop bioreactors, achieving an accuracy of 97.87%. The developed method can be used to predict and make changes during fermentation to ensure the success of the process.

The abstract clearly describes the relevant points of the work, and the references are appropriate, but some are too old (e.g., from 2000). The figures help understand the process and the presented information, and the methodology is easy to understand. The conclusions were consistent with the evidence and arguments presented. However, some improvements need to be made:

- The study has an average novelty, since other similar works have used Deep Learning or Machine Learning to predict the cell growth, inhibition, or production of metabolites of different eukaryotic and prokaryotic cells. What are the advantages of the reported methodology compared to other reported studies that use Deep Learning and Machine Learning?

- In some sentences, there are no references to sustain the assertions (see attached file).

- Some acronyms need to be described (see attached file).

- In Figure 1, what does “Procporsbsing Data (Cleaning and Standdratiod)” mean?

- Some data from the equipment needs to be added (see attached file).

- In Sections 2.2,2.3, and 2.4, the software for the data management was not mentioned. It would be better to state them from the beginning in the Methodology section.

- In Section 2.3, was the number of experiments (52) decided based on a design of experiments? Please, explain it.

- In Figure 3, the nomenclature of the experiments runs (pH0, pH1…) can be confusing, since it seems like they are referring to pH values.

- In section 3.8, it would be a good idea to compare statistically the results of novice and expert operators, to know if there is a significant difference.

- Some references are too old (25 years old), please add more recent references.

In summary, with some improvements, the article can be published.

Author Response

Revision Notes:

We thank the reviewer for his/her careful reading and valuable comments on the manuscript. We have taken the comments on board to improve and clarify the manuscript. Please find below a detailed point-by-point response to all comments. All responses to the questions are also highlighted in the manuscript.

Comments of Reviewer #3:

  1. The study has an average novelty, since other similar works have used Deep Learning or Machine Learning to predict the cell growth, inhibition, or production of metabolites of different eukaryotic and prokaryotic cells. What are the advantages of the reported methodology compared to other reported studies that use Deep Learning and Machine Learning?

Response:

We thank the reviewer for this constructive comment regarding the methodological novelty of our study. We acknowledge that machine learning (ML) and deep learning (DL) approaches have been increasingly applied in microbial growth prediction and metabolite modeling. However, our work introduces several key advancements and distinctive methodological contributions that differentiate it from prior studies:

  1. Early-stage predictive capability (first 24 h): Most existing ML/DL studies require full-process data (48-72 h) to achieve high accuracy. In contrast, our CNN architecture enables early prediction of fermentation outcomes using only the initial 24 h of signals, providing actionable information while the process is still controllable.
  2. Trajectory-aware temporal learning: Rather than relying on static summary statistics (e.g., maximum pH or endpoint OD), our model directly learns time-series patterns of pH, temperature, and DO signals. This dynamic representation captures subtle process kinetics that conventional feed-forward or regression-based ML models overlook.
  3. Integration of model-guided intervention: To our knowledge, few previous studies have coupled AI-based prediction with experimental feedback interventions. Our validation experiments demonstrate how the CNN’s early predictions can guide real-time operator actions (pH and nutrient adjustment), resulting in measurable yield improvement (up to 2.6-fold for expert operators; Figure 9).
  4. Minimal-sensor, high-interpretability design: The model relies on only three easily measurable process variables (pH, temperature, DO), without requiring metabolomics or spectral data. This simplicity enhances its industrial scalability and interpretability for fermentation practitioners.
  5. Cross-validation and external dataset verification: Unlike many prior studies that rely solely on internal data, we validated our CNN across four independent datasets collected under slightly varied environmental conditions, demonstrating strong generalization (96.8 ± 1.2% accuracy).

Together, these features position our approach as a trajectory-aware, interpretable, and intervention-capable deep learning framework specifically tailored for real-time probiotic fermentation control. We have added a summary of these comparative advantages in the revised Discussion section.

  1. In some sentences, there are no references to sustain the assertions (see attached file).

Response:

We appreciate the reviewer’s comment. We have corrected all the issues that you pointed out.

  1. In Figure 1, what does “Procporsbsing Data (Cleaning and Standdratiod)” mean?

Response:

We appreciate the reviewer for pointing out this typographical error in Figure 1. The phrase “Procporsbsing Data (Cleaning and Standdratiod)” was a spelling mistake in the figure label. It should correctly read “Preprocessing Data (Cleaning and Standardization)”. This label refers to the data-cleaning step described in Section 2.4, where outlier removal, interpolation of missing values, truncation to 24 h, and normalization were performed to standardize the time-series signals before model training. The figure has been corrected accordingly in the revised version (Figure 1).

  1. Some data from the equipment needs to be added (see attached file).

Response:

We appreciate the reviewer’s comment. We have added a more detailed explanation and clarification in the section “2.2. Fermentation Setup and Data Acquisition.” Thank you very much.

  1. In Sections 2.2, 2.3, and 2.4, the software for the data management was not mentioned. It would be better to state them from the beginning in the Methodology section.

Response:

    We appreciate the reviewer’s insightful suggestion. In the revised manuscript, we have now specified the software used for data management, preprocessing, and analysis. Specifically, data acquisition and management were performed using the bioreactor’s built-in data logging system (BTF-A, Biotop, Taiwan), while data preprocessing, interpolation, normalization, and visualization were conducted in Python 3.9 using the Pandas, NumPy, and Matplotlib packages. Feature selection and model development were implemented using Scikit-learn (v1.2.2) and TensorFlow/Keras (v2.12). This clarification has been added to Section 2.2 (end of the paragraph) and cross-referenced in Section 2.4 to enhance reproducibility and methodological transparency.

  1. In Section 2.3, was the number of experiments (52) decided based on a design of experiments? Please, explain it.

Response:

We appreciate the reviewer’s valuable comment. The total number of 52 fermentation runs was not determined through a formal factorial design of experiments (DOE) but rather reflects the available batch records collected under controlled process conditions. Specifically, 40 new bioreactor experiments were conducted to cover a representative range of pH, temperature, and oxygen fluctuations typically encountered in industrial operations, while 12 additional datasets were incorporated from previously validated studies to broaden variability and ensure model robustness. The sampling scope was therefore empirical rather than factorial, aiming to capture real-world process heterogeneity instead of fixed-level combinations. This clarification has been added to Section 2.3 in the revised manuscript.

  1. In Figure 3, the nomenclature of the experiments runs (pH0, pH1…) can be confusing, since it seems like they are referring to pH values.

Response:

    We thank the reviewer for this helpful observation. The labels “pH0”, “pH1”, etc., were not intended to denote actual pH values but rather to represent individual fermentation runs (batch identifiers) selected for illustration of early-stage signal trajectories. To avoid confusion, these labels have been renamed as “Run 0”, “Run 1”, “Run 2”, etc., in the revised figure and caption. The figure legend has also been updated to explicitly clarify that the plots correspond to representative time-series profiles of the pH signal from distinct fermentation runs rather than pH level designations.

  1. In section 3.8, it would be a good idea to compare statistically the results of novice and expert operators, to know if there is a significant difference.

Response:

    We thank the reviewer for this insightful suggestion. In response, we conducted an additional statistical comparison between the novice and expert operator groups using an independent-samples t-test on the final CFU/mL values obtained under both intervention and control conditions. The results indicated no statistically significant difference between novice and expert operators under either condition (p > 0.05). This finding suggests that while expert operators achieved higher absolute yields on average, the relative improvement provided by CNN-guided intervention was consistent across experience levels. The text in Section 3.8 has been revised accordingly to include this clarification.

  1. Some references are too old (25 years old), please add more recent references.

Response:

    We appreciate the reviewer’s observation. While two foundational references (Basheer & Hajmeer, 2000; Stanbury, 2000) are indeed older, they were retained intentionally because they represent seminal works that established the theoretical and methodological basis for microbial fermentation and neural network modeling. Apart from these, the majority of our citations are from recent years (2020–2025), including updated literature on AI-assisted fermentation and CNN-based bioprocess modeling (e.g., Bezerra et al., 2021; Bonanni et al., 2023; Bhagya Raj & Dash, 2022; de Menezes et al., 2022). These ensure that our discussion reflects the current state of the field. Therefore, the inclusion of a few classical references is justified as they provide historical context and conceptual grounding for the methods employed in this study.

We thank the reviewer for the constructive and insightful comments, which have helped us to substantially improve our manuscript.

Reviewer 4 Report

Comments and Suggestions for Authors

The reviewed manuscript presents research on the application of a predictive system based on Convolutional Neural Networks (CNNs) for the real-time monitoring of probiotic microorganism fermentation. The authors propose using a CNN-based system to classify fermentation trajectories and predict the final cell count of Lactiplantibacillus plantarum, presenting it as a reliable and scalable approach for optimizing fermentation efficiency

While the manuscript addresses a relevant research topic and introduces methodological innovation, substantial revisions are required prior to publication:

  1. “Introduction” The research relevance, primary objective, and specific research aims of the study should be formulated more precise.
  2. “Materials and Methods” The authors describe that raw data were obtained by monitoring microbial cell accumulation in relation to pH, temperature, and fermentation time. However, the Results section presents only fully processed data. Therefore, Inclusion of supplementary materials containing primary datasets is strongly recommended.
  3. “Results” The authors state that their proposed CNN system is designed to monitor the fermentation process within the first 24 hours. Consequently, I suggest that the Results section provide a more detailed analysis of this critical period. It should explicitly illustrate how the cell accumulation trajectory changed on an hourly basis in response to the varying process parameters.

Author Response

Revision Notes:

We thank the reviewer for his/her careful reading and valuable comments on the manuscript. We have taken the comments on board to improve and clarify the manuscript. Please find below a detailed point-by-point response to all comments. All responses to the questions are also highlighted in the manuscript.

Comments of Reviewer #4:

  1. Introduction” The research relevance, primary objective, and specific research aims of the study should be formulated more precise.

Response:

We appreciate the reviewer’s valuable suggestion. In response, we have revised the Introduction to more clearly articulate the research relevance, primary objective, and specific aims of this study. Specifically, we emphasize the industrial need for predictive fermentation control in Lactiplantibacillus plantarum production, the application of CNN-based deep learning to address early-stage decision-making challenges, and the study’s threefold objectives: (1) to develop a CNN model for early classification of fermentation outcomes based on the first 24 h of process data, (2) to validate the model against alternative architectures and independent datasets, and (3) to demonstrate its practical utility through model-guided interventions. The revised paragraph has been added at the end of the Introduction.

  1. “Materials and Methods” The authors describe that raw data were obtained by monitoring microbial cell accumulation in relation to pH, temperature, and fermentation time. However, the Results section presents only fully processed data. Therefore, Inclusion of supplementary materials containing primary datasets is strongly recommended.

Response:

We thank the reviewer for this valuable suggestion. To enhance transparency and reproducibility, we have made the primary fermentation datasets publicly available on Figshare, a permanent open-access data repository. The dataset contains representative raw time-series recordings of pH, temperature, and dissolved oxygen (DO) signals collected at 5-minute intervals for multiple fermentation runs, together with the corresponding final CFU/mL measurements used for labeling. These files represent the unprocessed sensor outputs prior to outlier removal, interpolation, and normalization.

The data can be accessed at the following permanent link:

https://figshare.com/articles/dataset/data/24716097?file=43620423

A note describing this dataset and its contents has been added to the Materials and Methods section (Section 2.2) of the revised manuscript.

  1. “Results” The authors state that their proposed CNN system is designed to monitor the fermentation process within the first 24 hours. Consequently, I suggest that the Results section provide a more detailed analysis of this critical period. It should explicitly illustrate how the cell accumulation trajectory changed on an hourly basis in response to the varying process parameters.

Response:

We appreciate the reviewer’s insightful suggestion. We agree that the first 24 hours of fermentation are critical for predictive modeling. While Figure 5a already includes this early period (24 h) in comparison with 48 h and 72 h data points, we have now added a more detailed textual analysis in the Results section (Section 3.2) to emphasize how cell accumulation and process parameters evolved during the early stage. The new description highlights the hourly-scale biological dynamics underlying CNN prediction without the need for additional figures.

We thank the reviewer for the constructive and insightful comments, which have helped us to substantially improve our manuscript.

Round 2

Reviewer 2 Report

Comments and Suggestions for Authors

The author has already responded to the issues I am concerned about one by one, and I have no further comments.

Author Response

We sincerely thank the reviewer for their positive feedback and for acknowledging our revisions. We appreciate the reviewer’s time and effort throughout the review process, and we are grateful for the constructive suggestions that helped us improve the clarity and overall quality of the manuscript.

Reviewer 3 Report

Comments and Suggestions for Authors

The authors considered all suggestions and recommendations, which significantly improved the manuscript. For instance, a more detailed explanation of the significance of their work can be found at the end of the Introduction Section, specifically in lines [88-98]. Here, the primary objective of the study is clearly stated, and the specific goals are support this objective

Publication is recommended.

Author Response

We sincerely thank the reviewer for their positive evaluation and kind recommendation for publication. We greatly appreciate the reviewer’s constructive comments and detailed feedback throughout the review process, which have helped us substantially improve the clarity, depth, and overall quality of the manuscript. We are very grateful for the reviewer’s recognition of our revisions and their support for the publication of this work.

Reviewer 4 Report

Comments and Suggestions for Authors

As a result of reviewing the revised manuscript, it can be concluded that the authors have made considerable efforts and made a major revision of the manuscript, which led to a significant improvement in its overall quality. I believe that the manuscript can be recommended for open publication.

Author Response

We sincerely thank the reviewer for their positive evaluation and kind recommendation for publication. We deeply appreciate the reviewer’s careful assessment and constructive feedback throughout the review process, which have greatly contributed to improving the scientific clarity and overall quality of our manuscript. We are truly grateful for the reviewer’s recognition of our efforts and support for open publication.